# Anti-Type II Diabetic Effects of Coix Seed Prolamin Hydrolysates: Physiological and Transcriptomic Analyses

**DOI:** 10.3390/foods13142203

**Published:** 2024-07-12

**Authors:** Guifang Zhang, Zhiming Li, Shu Zhang, Lu Bai, Hangqing Zhou, Dongjie Zhang

**Affiliations:** 1National Coarse Cereals Engineering Research Center, Heilongjiang Bayi Agricultural University, Daqing 163319, China; zgj2002@163.com (G.Z.); lizhiming1998@126.com (Z.L.); zshu996@163.com (S.Z.); bailu980618@126.com (L.B.); 18249175667@163.com (H.Z.); 2Food College, Heilongjiang Bayi Agricultural University, Daqing 163319, China; 3Key Laboratory of Agro-Products Processing and Quality Safety of Heilongjiang Province, Daqing 163319, China

**Keywords:** coix seed prolamin, hydrolysates, type 2 diabetes mellitus, transcriptomics

## Abstract

Previous studies have demonstrated that enzymatically prepared coix seed prolamin hydrolysates (CHPs) contain several bioactive peptides that efficiently inhibit the activity of target enzymes (α-glucosidase and dipeptidyl kinase-IV) in type 2 diabetes mellitus (T2DM). However, the anti-T2DM effects and potential mechanisms of CHPs as a whole in vivo have not yet been systematically explored. Therefore, we evaluated the preventive, therapeutic, and modifying effects of CHPs on T2DM by combining physiological and liver transcriptomics with a T2DM mouse model. The results showed that sustained high-fructose intake led to prediabetic symptoms in mice, with abnormal fluctuations in blood glucose and blood lipid levels. Intervention with CPHs effectively prevented weight loss; regulated abnormal changes in blood glucose; improved impaired glucose tolerance; inhibited the abnormal expression of total cholesterol, triglycerides, and low-density lipoproteins; alleviated insulin resistance; and restored pancreatic islet tissue function in mice fed a high-fructose diet. In addition, we found that CHPs also play a palliative role in the loss of liver function and protect various organ tissues (including the liver, kidneys, pancreas, and heart), and are effective in preventing damage to the liver and pancreatic islet cells. We also found that the intake of CHPs reversed the abnormally altered hepatic gene profile in model mice and identified 381 differentially expressed genes that could serve as key genes for preventing the development of T2DM, which are highly correlated with multiple glycolipid metabolic pathways. We demonstrated that CHPs play a positive role in the normal functioning of the insulin signalling pathway dominated by the IRS-1/PI3K/AKT (insulin receptor substrates-1/phosphoinositide 3-kinase/protein kinase B) pathway. In summary, CHPs can be used as effective food-borne glucose-modifying components of healthy foods.

## 1. Introduction

Diabetes mellitus is a serious chronic disease characterised by persistent hyperglycaemia and disorders of insulin secretion. Globally, it is recognised as a metabolic disease of public health significance. Diabetes mellitus is classified into three types: type I, type II and gestational diabetes mellitus. Type II diabetes mellitus (T2DM) is the most prevalent type, accounting for more than 90% of patients with the disease [1]. According to the reported diagnostic criteria for diabetes mellitus, three stages of T2DM development can be distinguished, namely impaired glucose tolerance (IGT), impaired fasting glucose (IFG), and overt diabetes [2]. Recent studies have demonstrated that the 5-year probability of IGT and IFG progressing to T2DM is 26% and 50%, respectively [2]. Therefore, early prevention and control of T2DM will help to delay or prevent the onset of the overt disease. Existing epidemiological, experimental, and clinical studies have demonstrated that daily dietary intake of a certain amount of healthy products, such as mixed grains, fruits, and vegetables, plays an important role in the prevention and control of T2DM [3]. The effect of these food substances are largely dependent on the rich bioactive factors, including proteins, peptides, dietary fibre and phenolic compounds in natural dietary products [4]. Therefore, the discovery, development, and utilisation of functional dietary factors with blood glucose-regulating activity has become a crucial in the glucose-lowering food industry and the field of precision nutrition.

Coix seed (*Coix lacryma-jobi* L.), also known as adlay, Job’s tears, or pearl barley, is a cereal belonging to the grass family, widely cultivated in East Asia. Coix seed is rich in a variety of biologically active substances, including polysaccharides, proteins, and oils, which are the building blocks for its multiple nutritional effects [5]. Coix seed protein is the main component of coix seed, accounting for 16–17% of the total nutritional content [5]. The distribution of essential amino acids of coix seed protein is superior to that of rice and other grains, and its leucine content is higher than what is found in *Cordyceps sinensis*, which is an excellent blood sugar-regulating ingredient [6]. Recent studies have highlighted that coix seed protein has significant effects in the control and treatment of T2DM [7], and release bioactive peptides with higher hypoglycemic activity through bio-enzymatic and fermentation techniques, thus lending credence to the notion that further research into this substance is worth exploring. However, the efficacy of coix seed protein and its derivative bioactive peptides in regulating T2DM have not yet been comprehensively evaluated. As a result, their development potential has not been adequately assessed. One reason for this is the prolonged lack of interest in this raw material. Therefore, in the past 2 years, our research group has studied the therapeutic and regulatory effects of coix seed prolamin peptides in T2DM using coix seed prolamin (which accounts for 44.74% of the total nutritional value of coix seed protein [6]) as a raw material. We have previously reported that coix seed prolamin peptides obtained by the bio-enzymatic method can regulate blood glucose by inhibiting multiple targets (α-glucosidase, dipeptidyl kinase-IV). We have further explored the conformational relationship between coix seed prolamin peptides and the target enzymes via mass spectrometry, bioinformatics analysis, and in vitro validation, and examined their bioavailability properties [8,9,10,11]. These findings confirm that coix seed prolamin peptides are potential food adjuvants for the modulation of T2DM and deserve further investigation. However, from the perspective of food applications, it is not clear whether the coix seed prolamin hydrolysates (CHPs) of the mixed peptide system can exert the corresponding glucose-regulating efficacy in vivo. There is also a need to investigate potential pathways for the prevention or modulation of T2DM.

With the improvement of living standards, fructose is widely used as a sweetener in the food industry, which leads to a large amount of intake because of its low price, good taste, and low satiety. High fructose intake induces insulin resistance. Metabolism of fructose in the liver leads to high lipid production, elevated free fatty acids, and impaired insulin signalling receptors, which reduce tissue response to insulin signalling [12]. In addition, fat from fructose metabolism accumulates in the liver, causing non-alcoholic fatty liver disease, which affects the normal function of the liver and ultimately produces insulin resistance, in which hepatic fructokinase plays a key role in the development of insulin resistance [13,14]. Therefore, it is scientifically and practically important to examine the regulatory role of CHPs on T2DM (typically characterised by dysglycemia and insulin resistance) induced by high-fructose diets.

Therefore, our study aims to investigate the effects of CHPs on the development of T2DM by utilising a T2DM mouse model with a continuous high-fructose diet. We also aim to evaluate the preventive and therapeutic effects of CHPs on T2DM from a physiological and biochemical perspective and to explore the potential mechanism of CHPs against T2DM at the genetic level using liver transcriptomics.

## 2. Materials and Methods

### 2.1. Materials and Main Reagents

The preparation method of coix seed prolamin was conducted by referring to the previous report of the group [8]. The 60% high-fructose feeds and blank control group feeds were purchased from Jiangsu Xietong Pharmaceutical Bio-engineering Co., Ltd., Nanjing, China. Acarbose (Met, Positive drug) was purchased from Lepu Medical Technology (Beijing) Co., Ltd., Beijing, China. A glucose meter was purchased from Sannuo Biosensor Co., Ltd., Changsha, China. Detection kits of insulin (FINS), total cholesterol (TC), triacylglycerol (TG), high-density lipoprotein (HDL-c), low-density lipoprotein (LDL-c), alanine aminotransferase (ALS), and aspartate transaminase (AST) were purchased from Shanghai Enzyme-linked Biotechnology Co., Ltd., Shanghai, China.

### 2.2. Preparation of CHPs

We utilised the method reported by Li et al. in the previous panel for the preparation of CHPs [8]. First, a certain amount of coix seed prolamin sample was prepared and added into a 4% solution mixed with deionised water and sequentially pre-treated with heat (90 °C for 30 min) and ultrasonication (300 W at 40 kHz for 30 min). Subsequently, the mixed solution was placed in a magnetic stirring pot, and the temperature was maintained at 50 °C while the pH of the solution was adjusted to 9.0. Under continuous stirring, 8000 U/g of alkaline protease was added to start the enzymatic hydrolysis. After 2 h of hydrolysis, the hydrolysed products were rapidly inactivated by placing a water bath at 90 °C for 20 min, followed by centrifugation at 3500× *g* for 20 min and freeze-drying of the supernatant to obtain CHPs.

### 2.3. Design of the Experimental Animal

All experiments were designed in strict accordance with the Regulations for the Administration of Laboratory Animals and were approved by the Ethics Committee of Animal Experiments of Heilongjiang Bayi Agricultural University, China. Seventy-two C57BL/6J mice (age 6–8 weeks) were purchased from Harbin Medical University (Daqing), China (laboratory animal production licence: SCXK (Hei) 2019-001). They were kept at a feeding temperature of 25 ± 2 °C, relative humidity of 50 ± 5%, and day/night cycle of 12 h/12 h. The control group was fed adaptively for 1 week before the start of the experiment and then randomly divided into six groups according to body weight. The test experiment commenced with 12 animals in each group, and different feeding and intervention modes were utilised in various groups. The feeding details are as follows: normal group (NC): normal feed + saline; model group (DC): 60% high-fructose feed + saline; positive control group (Met): 60% high-fructose feed + acarbose; low-dose intervention group (CHPs-L): 60% high-fructose feed + 125 mg/kg BW CHPs; medium-dose intervention group (CHPs-M): 60% high-fructose feed + 250 mg/kg BW CHPs; high-dose intervention group (CHPs-H): 60% high-fructose feed + 500 mg/kg BW CHPs. Different amounts of CHPs were dissolved in saline by gavage (2 mL) for 11 weeks, and the specific experimental design is shown in Figure 1A. Blood samples of each group were collected from tail vein, and the blood glucose levels were measured using a glucometer according to the manufacturer’s instructions (Sannuo Biosensor Co., Ltd., Changsha, China). After 10 and 11 weeks of intervention, the blood glucose levels of mice in the DC group were observed to be greater than 6.1 mmol/L for two consecutive weeks. This meets the clinical diagnostic criteria of “prodiabetes”. Also, the metabolic symptom of glycaemic imbalance has been induced successfully by the dietary intervention [2]. At the end of the experiment, mice were euthanised by cervical dislocation after a 12 h fast. Blood samples were rapidly collected and centrifuged at 1000× *g* for 10 min to assess serum biochemical indices. Also, the liver, pancreas, kidneys, and heart were cleaned and weighed, and samples from the liver and pancreas were fixed with 4% paraformaldehyde and then stained with HE (haematoxylin and eosin) for histopathological observation. In addition, a portion of the liver was snap-frozen in liquid nitrogen and stored at −80 °C for transcriptomic analysis.

### 2.4. Oral Glucose Tolerance Test (OGTT)

OGTT was performed in the last week of CHP therapy. After overnight fasting, the mice a 2 g/kg body-weight glucose solution, followed by blood sampling from the tail vein using a glucometer to determine blood glucose levels at 0, 30, 60, 90, 120, and 180 min.

### 2.5. Serum Biochemical Analysis

Serum levels of FINS, TC, TG, HDL-c, LDL-c, AST, and ALT were measured by ELISA kits (Shanghai Enzyme-linked Biotechnology Co., Ltd., Shanghai, China), and the procedure was described in the kit instructions. Additionally, we applied Atherogenic Index of Plasma (AIP) = lg (TG/HDL-c), Castelli’s Risk Index I (CRI-I) = TC/HDL, and Castelli’s Risk Index II (CRI-II) = LDL-c/HDL-c.

### 2.6. Assessment of Insulin Resistance and Islet Cell Function

Insulin resistance (HOMA-IR), β-cell function (HOMA-β), and insulin sensitivity index (ISI) were calculated based on measured FBG and FINS. The calculation metrics were HOMA-IR = (FBG + FINS)/22.5, HOMA-β = 20 × FINS/(FBG-3.5), and ISI = LN [(FBG × FINS)^−1^], respectively.

### 2.7. Histology Analysis of the Liver and Pancreas

HE staining sectioning was performed on liver and pancreas tissues that had been fixed, and the brief operation was as follows: (1) tissue decolourisation, (2) tissue transparency, (3) wax dipping, (4) embedding, (5) slicing and baking, (6) slice deparaffinisation, (7) haematoxylin staining, (8) differentiation in differentiation solution, (9) eosin staining, (10) decolourisation, (11) transparency, and (12) sealing. The prepared slides were observed under a microscope.

### 2.8. Transcriptomic Analysis of Mouse Liver under CHP Intervention

#### 2.8.1. RNA Extraction, Library Construction, and Sequencing

The hepatic tissues of five mice in the NC, DC, and CHP groups were subjected to transcriptomic analysis. The extracted RNA was subjected to agarose gel electrophoresis and nanodrop manipulation to analyse the integrity of the RNA in the samples and assess its concentration and purity using the optical density (OD) 260/280 and 260/230 ratios, respectively. After the samples were tested, Oligo(dT) beads were used to isolate the mRNA. The mRNA was fragmented by adding a fragmentation reagent, and the first cDNA strand was synthesised using second-strand synthesis mix, followed by end polishing and the addition of “A”. Subsequently, the RNA-Seq adapter was connected, and the ligated product underwent purification for PCR amplification. Magnetic beads were employed to purify and recover target products ranging from 300 bp to 600 bp for sequencing. The effective concentration of the library was subsequently quantified using Qubit and Q-PCR methods (with an effective concentration > 20 nm). After passing library inspection, libraries with varying concentrations and downstream data were pooled, then sequenced using the Illumina Novaseq 6000 platform with PE150 sequencing, based on the principle of sequencing by synthesis.

#### 2.8.2. Quality Control and Standardised Analysis of Sequencing Data

Raw images generated from high-throughput sequencing, such as Illumina PE150/SE50, were initially processed into sequenced reads using CASAVA base recognition and saved in FASTQ format. The sequencing data were filtered, which involved removing reads containing adapters and those with an N ratio greater than 0.002 (N indicates that the base information cannot be determined). This step aimed to analyse the distribution of the sequencing error rate, ensuring that the accuracy of base identification surpassed 99%, while also examining the GC proportion within nucleotide sequences. Following these quality checks, clean reads were acquired for subsequent analyses. Subsequently, the clean reads were aligned to the mouse genome (GRCm39) annotated in the Ensemble database (Mus_musculus.GRCm39.109.gtf) using HISAT2 software (version 2.2.1) with default parameters. This alignment ensured that the proportion of reads (total mapped) aligned to the reference sequence was higher than 80%, and the number of reads with multiple mapped positions on the reference sequence (multiple mapped reads) was less than 20%. After completing the alignment, transcripts obtained from the splicing of each sample were merged using Stringtie software (https://ccb.jhu.edu/software/stringtie/, version number: v2.2.0, accessed on 30 August 2023) to identify novel transcripts.

#### 2.8.3. Screening for Differential Genes

Gene levels in the transcriptome were quantified using the HTSeq software package (http://htseq.readthedocs.io/en/release_0.9.1, accessed on 30 August 2023, version number: 0.9.1), and gene expression values were expressed as fragments per kilobase of transcript sequence per million base pairs sequenced (FPKM). The edgeR software (http://www.bioconductor.org/packages/release/bioc/html/edgeR.html, accessed on 30 August 2023, version number: 3.19) was employed for analysing the significance of gene expression difference, with a padj value less than 0.05 utilised as the threshold of statistical significance. If the number of differences identified with padj less than 0.05 was insufficient, additional screening was conducted using a *p*-value less than 0.05. The aforementioned liver transcriptomic analysis was performed by Hangzhou Kaitai Biotechnology Co. Ltd., Hangzhou, China.

#### 2.8.4. Gene Ontology (GO) and Kyoto Encyclopaedia of Genomes (KEGG) Enrichment Analysis

Pathway enrichment analysis of GO and KEGG for differential genes was performed using clusterProfiler software (http://www.bioconductor.org/packages/release/bioc/html/clusterProfiler.html, accessed on 30 August 2023, version number: 3.19). Both GO and KEGG pathways were significantly enriched (*p* < 0.05).

### 2.9. Statistical Analysis

The data on physiological and biochemical indices obtained in this study were expressed as mean ± standard deviation (SD) and were analysed for variability and plotted using GraphPad Prism 8.4.3 software. Analysis of significant differences between groups (compared with the DC group) included one-way ANOVA and Tukey’s multiple comparison test. The statistical significance of the results was configured as *, *p* < 0.05; **, *p* < 0.01; ***, *p* < 0.001; ****, *p* < 0.0001.

## 3. Results

### 3.1. Effect of CHPs on Body Weight and Organ Index in Mice

The sustained intake of a high-fructose diet induces metabolic syndrome in mouse models, initially manifesting as disorders of glucose and lipid metabolism, followed by the progression to chronic metabolic diseases such as insulin resistance, hyperuricaemia, hyperglycaemia, and non-alcoholic fatty liver disease [15]. As shown in Figure 1B, a dietary pattern high in fructose resulted in slower weight gain in mice compared to the NC group. The body weights of the mice in the high-fructose diet group at the end of the experiment were lower than those of the NC group (Figure 1C), with a 20.23% decrease in the DC group compared to the NC group. However, the CHP-treated group exhibited less weight loss than the DC group (*p* < 0.05) (Figure 1C). These findings indicate that the high-fructose diet delayed normal development and limited body weight gain. Nevertheless, CHPs reduce this effect, attenuate wasting, and restore normal weight gain in mice. The findings are consistent with those reported by Zhu et al. [16] and Li et al. [17].

The organ index not only provides insight into organ pathology in mice on a continuous high-fructose diet, but also reflects the development of metabolic syndrome and its complications characterised by disorders of glucose–lipid metabolism [17]. The effects of CHPs on the liver, pancreas, kidney, and heart indices are shown in Figure 1D, Figure 1E, Figure 1F, and Figure 1G, respectively. Compared with the NC group, the DC group showed an increase in the liver and cardiac indices, and a decline in the kidney and pancreas indices, but there was no statistically significant difference (*p* > 0.05). This suggests that the organs of mice on the high-fructose diet showed a propensity for disease development, with a heightened risk of progressing to conditions such as diabetes, renal dysfunction, and hypertension. Following intervention with CHPs, the abnormal trend in each organ’s index slowed down and gradually approached that of the NC group, indicating that CHPs had a significant inhibitory effect on the malignant development of organs produced by high fructose. However, it is worth noting that there were no significant differences in liver, pancreas, and cardiac indices with the intervention of CHPs compared to DC (*p* > 0.05), except for the renal index.

### 3.2. Effect of CHPs on Glucose Management in Mice

FBG was continuously monitored in each group over the 11-week intervention period. The blood glucose values and the trend of blood glucose changes did not change significantly in each group of mice during the pre-feeding period (0–9 weeks). However, the blood glucose values of the DC group continued to increase in the later period (9–11 weeks) and reached the highest value after 11 weeks. This suggests that the high-fructose diet induced abnormal blood glucose levels in mice and that the simultaneous intake of CHPs played a positive role in suppressing the abnormal elevation of blood glucose levels (Figure 2A). In the OGTT experiment, blood glucose levels in the NC group remained low after gavage of the glucose solution and did increase substantially. Conversely, blood glucose in the DC group rose rapidly within 0–30 min of glucose solution administration and remained at a high level until the end of the experiment (Figure 2B). In contrast, blood glucose levels increased less rapidly in each group than in the DC group following glucose solution gavage (Figure 2B). The calculated area under the curve (AUC) for changes in blood glucose levels also confirmed that the AUC was significantly lower in the CHP group than in the DC group (*p* < 0.05) (Figure 2C). These results demonstrated that CHPs can protect mice from impaired glucose tolerance induced by a high-fructose diet, with the optimal effect of CHPs-H.

### 3.3. Effect of CHPs on Insulin Sensitivity in Mice

Continued high fructose intake triggers insulin-resistant effects in the body, mainly due to fructose metabolism in the liver, which generates large amounts of lipids and results in lipid peroxidation. This could ultimately lead to damaging insulin signalling receptors [12]. In our experiment, the high-fructose diet induced insulin resistance in mice, as evidenced by higher serum insulin concentrations in the DC group (9.75 ± 0.43 mIU/L) than in the NC group (7.77 ± 0.54 mIU/L) (*p* < 0.0001). The intake of CHPs helped regulate abnormally elevated insulin levels, with CHPs-H showing the most effective reduction, by 14.28% compared to DC, albeit lower than what is observed in the Met group (Figure 2D). After 11 weeks on the high-fructose diet, ISI decreased, HOMA-β decreased, and HOMA-IR increased in the DC group compared to mice in the NC group, but the indices returned to the NC level after the CHP intervention (Table 1). These findings demonstrate that CHPs can restore the function of pancreatic islet tissues to some extent and are resistant to islet cell loss and insulin resistance induced by a high-fructose diet.

### 3.4. Effect of CHPs on Serum Lipid Metabolism, ALT, AST, AIP, CRI-I and CRI-II in Mice

Metabolic syndrome caused by high-fructose diets is often accompanied by dyslipidaemia, which is generally characterised by disorders of lipid metabolism [18]. Compared with the NC group, mice in the DC group had elevated TC, TG, and LDL-c levels and decreased HDL-c levels in serum, but the intake of CHPs reversed this phenomenon and prevented lipid metabolism disorders (Figure 3A–D). CHPs prevented lipid metabolism disorders in mice in a dose-dependent manner, with CHPs-H showing the greatest improvement (*p* < 0.0001). In addition, the intervention of CHPs significantly ameliorated the abnormally altered AIP, CRI-I, and CRI-II (Figure 3G–I) in high-fructose dietary mice (*p* < 0.001). These results suggest that CHPs can alleviate the risk of cardiovascular morbidity induced by a high-fructose diet, accelerate cholesterol metabolism, and effectively treat dyslipidaemia.

The liver is the main site of glycolipid metabolism, and the fructose ingested in this study was mainly metabolised and converted in the liver, with excessive metabolic activity causing pressure [19]. The maintenance of normal levels of AST and ALT in the body is a marker of hepatocellular health and deranged levels are typical of hepatocellular injury [17]. As shown in Figure 3E,F, there was an upward trend in both AST and ALT in the DC group after a sustained high-fructose diet relative to the NC group, indicating that hepatocyte loss had been induced. However, CHP uptake prevented impaired liver functioning, with CHPs-H showing the best effect (*p* < 0.0001). This suggests that CHPs can effectively alleviate high-fructose-induced hepatic dysfunction and repair damaged hepatocytes.

### 3.5. Histopathology of the Liver and Pancreas

As shown in Figure 4A, the liver tissues of the mice in the NC group had normal texture, with a normal hepatic lobule structure and orderly hepatocyte arrangement, devoid of any no obvious pathological changes. Conversely, in the livers of mice in the DC group, hepatocyte damage was evident, characterised by a blurring of the liver lobules, a disorganised hepatocyte arrangement, hepatocyte ballooning, the presence of fat droplets, and infiltration with numerous inflammatory cells. CHP uptake resulted in a notable reduction in these histopathological changes along with improvement in the hepatic lobular structure, hepatocyte arrangement, congestion, oedema, and vacuolisation. CHPs effectively alleviated high-fructose diet-induced morphological lesions in mice.

As shown in Figure 4B, mice in the NC group exhibited more abundant cells that were neatly dispersed, organised, and structurally intact without rupture or shrinkage. Mild atrophy and apoptosis of pancreatic islet cells were observed in the DC group; however, the extent of this effect was limited. Following CHP uptake, islet morphology improved, cell damage was reduced, and the morphology and sorting of individual islet cells normalised. These findings suggest that the intake of CHPs could partially repair islet cell damage triggered by a high-fructose diet.

### 3.6. Transcriptomic Analysis of Mouse Liver

#### 3.6.1. Acquisition and Processing of Data

From the above analysis, it can be observed that the uptake of CHPs along with a sustained high-fructose diet can effectively prevent the pathogenesis of glycolipid metabolism-associated T2DM and provide a positive influence at the physiological and biochemical levels. To better understand the effects of CHPs on liver gene expression levels, mouse livers from the NC-, DC-, and CHP-treated groups were sequenced and analysed using transcriptomic technology. As shown in Table 2, a total of 36,089,996–43,072,784 raw reads were obtained for each sample after initial sequencing, and 35,695,896–43,421,540 clean reads were obtained after quality control. All samples had a comparison ratio greater than 80% (except NC_2), Q_20_ greater than 97%, Q_30_ greater than 93%, and a GC content of approximately 50%. These results indicate that the quantity and quality of the sequencing met the requirements for subsequent analysis, and the RNA-seq results obtained were reliable.

#### 3.6.2. Differential Gene Expression in NC, DC, and CHP Groups

As shown in Figure 5A, principal component analysis (PCA) was performed on the transcripts of the three groups, with the first principal component contributing 34.33% and the second principal component contributing 11.91%. The three groups were clearly differentiated in terms of each primary score, with the NC and CHP groups mainly concentrated on the left side and the DC group on the right side, whereas the NC and CHP groups were found to be in close proximity. The above results indicate that under the continuous high-fructose diet, gene expression in the mouse liver was abnormally changed compared with the normal dietary group, and the genetic expression observed in mice receiving CHPs was similar to the normal dietary group. The observation in Figure 5B also shows that the correlation between the NC and CHP groups was significantly higher than the correlation between the NC and DC groups, as evidenced by the colour depth of the DC group being lower than that of the other two groups. This suggests that the samples from the NC and CHP groups have a higher degree of similarity and the gene expression profiles are closer to each other.

Figure 5C and Figure 5D show the differential gene expression between the DC vs. NS and CHP vs. DC groups, respectively (*p* < 0.05, |log2FoldChange| > 1). Compared with the NC group, a total of 1134 genes were significantly differentially expressed in mice in the DC group, of which 647 were upregulated (mainly including Scd1, Aldob, Gstm1, Hba-a1, Fasn, Eno1, Cyp3a11, Me1, Gpi1, and Acly) and 487 were downregulated (mainly including Map3, Mgst1, Scp2, Phyh, Mup20, Mup17, Mup12, Serpina3k, and Serpina1e). A total of 719 genes differed between the CHP and the DC group, of which 450 were upregulated (mainly including Mup3, Mup20, Mup17, Mup12, Rn18s-rs5, Serpina3k, Apoa4, and Akr1c6) and 269 were downregulated genes (mainly including Ass1, Fasn, Acaa1b, Por, Acaca, Lgfbp1, and Cyp2a4). The hierarchical clustering heatmaps of differential genes drawn for the DC vs. NC and CHP vs. DC groups were also shown to discriminate effectively between the two groups (Figure 5C and Figure 5D, respectively), and an opposite trend was observed in the expression levels of the same differential genes between the two groups. An additional 381 differential genes were common to all three groups (Figure 5E), and these genes can be considered key regulatory genes for reversing or delaying the onset of diabetes following the use of CHPs. Figure 5F shows a heatmap of CHPs reversing these differential genes. The gene expression profiles of liver tissues in DC mice clearly differed from those of the NC and CHP groups. Notably, the highly expressed genes in DC were lowly expressed in NC and CHP groups, and vice versa (Figure 5F). These results indicate that the gene expression profile of T2DM mice can be significantly reversed by CHPs, ensuring a genetic expression profile similar to that of normal liver.

#### 3.6.3. GO and KEGG Analysis

Further GO annotation of the up- and downregulated genes under CHP treatment was performed to analyse and summarise the top 20 ranked biological pathways. Among the biological processes involved in differential gene upregulation following treatment with CHPs, the primary biological processes included negative regulation of peptidase activity, negative regulation of endopeptidase activity, xenobiotic metabolic processes, oestrogen metabolic processes, and amyloid-beta metabolic processes. The primary cellular components include the plasma membrane, extracellular space, endoplasmic reticulum, and extracellular region. The primary molecular functions included oxidoreductase, monooxygenase, serine-type endopeptidase inhibitor, and peptidase inhibitor activities (Figure 6A). Among the biological processes annotated as downregulated differential genes following treatment with CHPs, the primary biological processes included lipid metabolism, proteolysis, inflammatory response, positive regulation of T cell proliferation, and xenobiotic metabolism. The primary cellular components include the plasma membrane, extracellular space, extracellular region, and intracellular protein transport. The primary molecular functions include hydrolase, oxidoreductase, steroid hydroxylase and monooxygenase activities (Figure 6B).

The KEGG enrichment results showed that the differentially upregulated genes under CHP intervention were mainly enriched in complement and coagulation cascades, retinol metabolism, steroid hormone biosynthesis, metabolism of xenobiotics by cytochrome P450, cytochrome P450, and fatty acid degradation (Figure 6C). Downregulated differential genes were mainly enriched in pancreatic secretion, NF-kappa B signalling pathway, TNF signalling pathway, Toll-like receptor signalling pathway, arachidonic acid metabolism, and JAK-STAT signalling pathway (Figure 6D). Moreover, in addition to the pathways shown in Figure 6D, the downregulated differential genes were also significantly enriched (*p* < 0.05) in several key pathways associated with the onset and development of diabetes mellitus, namely MAPK, PI3K-Akt, FoxO, AMPK signalling pathways, and fatty acid biosynthesis. Regarding these enriched pathways, the cytochrome P450 pathway and the CYP epoxygenase-EET-sEH system play important roles in the onset and development of insulin resistance and can delay the development of insulin resistance caused by fructose dietary intake by upregulating genes such as Cyp2j3, Cyp2e1, and Cyp1a2 [20,21]. Additionally, in the intervention of certain active dietary factors on T2DM, the fatty acid degradation pathway emerges as an important regulatory pathway, which can effectively control T2DM by addressing symptoms of glycolipid metabolism disorders [22,23]. Furthermore, numerous studies have shown that the NF-kappa B signalling pathway, TNF signalling pathway, Toll-like receptor signalling pathway, arachidonic acid metabolism, and JAK-STAT signalling pathway are closely associated with the development of T2DM [24]. In conclusion, CHPs achieve T2DM resistance through multiple pathways, making a multi-target, multi-pathway food-borne active hypoglycaemic substance.

#### 3.6.4. Analysis of Relevant Differentially Expressed Genes in the Insulin Signalling Pathway

Our findings suggest that CHP regulates liver-related pathways to exert anti-T2DM effects. Considering the experimental phenomena and purpose of the investigation, we focused on the insulin signalling pathway, which is closely related to the development of T2DM. As shown in Table 3, the key genes involved in this signalling pathway have been identified, including genes mediating the upstream protein of the PI3K/AKT signalling pathway, such as Socs1, Socs3, Irs1, Pik3r1, Pdpk1, and Akt1, and those from glucose-transporting protein family GLUT, namely Slc2a4 and Slc2a2. The key genes involved in glycogen synthesis include Gsk3b, Ppp1ca, Pygl, and Pygm, while the key genes involved in glycolysis/glycolysis include Gck, FBP1, G6pc3, and Pck1. The genes involved in fatty acid synthesis include Srebf1, Acaca, and Fasn. One notable feature was that the expression levels of these genes deviated from the normal state and were up- or downregulated after high-fructose dietary feeding (DC group) relative to the NC group. This suggests that CHPs can influence extracellular glucose uptake and absorption, glycogen synthesis, glycolysis, and fatty acid synthesis in the mouse liver by mediating the insulin signalling pathway to maintain its normal function and ultimately exert the antidiabetic effect in T2DM mice.

Insulin signalling is triggered by the binding of insulin to the insulin receptor on the target cell membrane. Insulin receptor substrates that have been activated activate downstream signalling pathways to perform the biological functions of insulin. Among them, the PI3K-Akt pathway is one of the important pathways for insulin to fulfil its biological functions [25]. The mediating pathways can be roughly summarised as follows [26]: when extracellular insulin signalling is delivered to the cell via IR and IRS, the SH2 domain of the regulatory subunit of PI3K, p85, will bind to the phosphorylated IRS, thereby activating the catalytic subunit of PI3K, p110. In response to growth hormone stimulation, the p110 subunit catalyses phosphatidylinositol (4,5)-bisphosphate (PIP2) into phosphoinositide (3,4,5)-triphosphate (PIP3), and PIP3 ultimately activates AKT and other downstream factors. PIP3 binds to the PH structural domain of AKT, thereby promoting AKT phosphorylation by upstream kinases. First, 3-phosphoinositide-dependent protein kinase-1, (PDK1) phosphorylates the catalytic structural domain of AKT, Thr308, and thus 10% of AKT activity is activated. Then, the mammalian target of rapamycin complex 2 (mTORC2), DNA-dependent protein kinase (DNA-PK), and ataxia telangiectasia mutated kinase (ATM) phosphorylate the AKT regulatory domain Ser473, thereby fully activating the kinase activity of AKT. Activated AKT is released from the plasma membrane and translocates to the cytoplasm, mitochondria, and nucleus to phosphorylate various substrates. Important substrates of AKT are glycogen synthase kinase 3β (GSK3β), forkhead box protein O1 (FOXO1), and protein kinase B substrate 160 (AKT substrate 160, AS160), which regulate glycogen synthesis, gluconeogenesis, and glucose uptake, respectively.

## 4. Discussion

In our present study, mice with sustained high fructose intake exhibited a reduced rate of body weight gain and abnormal fluctuations in fasting blood glucose, especially in the later stages (9–11 weeks) of our intervention. Weight loss and sustained high levels of blood glucose (>6.1 mmol/L), which are consistent with prodromal diabetes, were noted in this period. Previous studies have reported that excessive catabolism and utilisation of body fat and proteins in mice with progressive disease can lead to impaired weight gain and weight loss due to insufficient glucose utilisation [27]. Additionally, at the end of the intervention, the glucose tolerance of the mice was weakened, as evidenced by consistently higher blood glucose levels in the DC group than in the NC group observed during OGTT (0–120 min), and significantly higher ACU than in the NC group (*p* < 0.0001). An important cause of this impaired glucose metabolism is inadequate insulin secretion and decreased sensitivity in mice [28]. HOMA-IR, HOMA-β, and ISI were utilised to assess the degree of peripheral insulin resistance and islet cell function [29]. Our results showed that DC mice had abnormalities in the relevant indices of insulin metabolism (including FINS, HOMA-IR, HOMA-β, and ISI), and islet cell shrinkage and reduced number were also found in HE staining of the pancreas, resulting in insulin resistance. Following intervention, CHP mice showed improved body weight loss, enhanced glycaemic control, and alleviated symptoms of impaired insulin secretion and insulin resistance. Similar research involving protein hydrolysate interventions in T2DM have yielded significant results, such as Wang et al. who fed walnut protein hydrolysate (3–10 kDa) at concentrations of 200, 500, and 800 mg/kg/d orally to T2DM mice (*n* = 10) and reported that after 4 weeks, the mice showed a 64.82% reduction in fasting glucose levels and a 23.71% increase in insulin secretion [30]. Yan et al. force-fed *ICR* mice (*n* = 6) with wheat germ protein hydrolysate (>5 kDa, 3.24 mg/mL) at a dose of 48.6 mg/(kg·d), and this led to an alleviation of polydipsia, polyphagia, and hyperglycaemia after 2 weeks, with an effective control of postprandial glycaemia [31]. In these cases, it can be determined that most of the food-borne peptides used for in vivo experiments are mixed peptide. Most of these hydrolysates are obtained through simple treatments such as desalination and decolourisation. This in fact fits the application scenario of food-borne peptides in dietary nutrition and health interventions and has the advantage of low costs. In the future, this product form may be the direction of research and commercialisation of coix seed prolamin peptides.

The development of T2DM is accompanied by symptoms of dysregulation of lipid metabolism, which are positively correlated with the development of insulin resistance [32]. This condition is characterised by elevated TC, TG, and LDL-c, and decreased HDL-c levels. In this study, CHPs significantly reduced already dysregulated TC, TG, and LDL-c levels while elevating HDL-c levels, thereby improving impaired lipid metabolism, with CHPs-H demonstrating a superior dose-dependent effect. Meanwhile, CHPs significantly ameliorated imbalanced AIP, CRI-I, and CRI-II, which was beneficial in reducing the risk of cardiovascular disease development in mice on high-fructose diets. An obvious feature of dysregulated lipid metabolism is the tendency to form excessive lipid deposition in the liver, leading to lipid peroxidation and an excessive production of free radicals in the body, which ultimately results in oxidative stress [33]. In this study, CHPs reduced ALT and AST levels, restored basal liver function, and reversed hepatocyte loss. Results of HE-stained liver sections also show that the intervention with CHPs significantly reduces the accumulation of liver lipids and alleviates the degree of steatosis, which has a good protective effect on the mouse liver. Numerous studies have confirmed that elevated lipid levels are an important cause of cardiovascular disease, and regulation of lipid metabolism can prevent cardiovascular disease to a greater extent [34]. Moreover, all organ indices were altered when mice were on the high-fructose diet; however, only the renal index significantly differed when compared to NC (*p* > 0.05). Continued disruption of organ indices is known to lead to organ pathology; however, CHPs have mitigated this progression. Consistent studies have shown that food-derived protein peptide intake, such as soy protein peptides [35], ginseng peptides [36], and whey protein peptides [37], improve lipid abnormalities and restores metabolic disorders in T2DM model.

As the “metabolic centre” of the human body, the liver is crucial for maintaining normal glucose metabolism and is an important organ for glucose regulation, as physiological activities such as glycogen synthesis, glycolysis/glycolysis, and fatty acid synthesis/catabolism occur [38]. Following the analysis of the livers of the NC, DC, and CHP groups by RNA-seq, PCA, and correlation analyses, we found that the samples of the DC group were separated from those of the other two groups, while those of the NC and CHP groups were clustered together and had a higher degree of correlation. Regarding the expression levels of differential genes, NC and CHP groups showed similar findings, while the upregulation/downregulation trends of various genes in the DC group were in contrast to those of the other two groups, thus confirming that CHPs reversed the aberrant changes in relevant genes during the development of T2DM. CHP- and DC-enriched pathways include several pathways linked to glycolipid metabolism, such as NF-kappa B signalling pathway, arachidonic acid metabolism, JAK-STAT signalling pathway, MAPK signalling pathway, PI3K-Akt signalling pathway, FoxO signalling pathway, fatty acid biosynthesis, and AMPK signalling pathway. These pathways were consistent with those reported to be involved in T2DM [39,40,41].

Focusing on the insulin signalling pathway, it was found that the gene expression levels of key protein signalling could be restored to use, compared to those of the NC group after the effects of CHPs, including the pathways involved in fatty acid synthesis, glycolysis, and gluconeogenesis. Among them, the suppressors of cytokine signalling (gene name: Socs1 and Socs3) appeared to be downregulated in the DC group, while the expression level of mRNAs appeared to be upregulated after intervention with CHPs. The two protein molecules interacted with the insulin receptor, and their overexpression negatively regulated an insulin-induced phosphorylation of insulin receptor substrates (IRS, gene name: Irs1 and Irs2), which is one of the most important factors contributing to insulin resistance in the body [42,43]. Meanwhile, Irs1, serving as a docking site for activating downstream insulin signalling, exhibited elevated expression levels under the influence of CHPs. Additionally, phosphoinositide 3-kinase (PI3K, gene name: Pik3r1) as a key upstream mediator, achieved a resurgence in gene expression levels in response to CHP intervention. This suggests that CHPs effectively mediate the upstream pathway of the insulin signalling pathway, ensuring efficient signalling [44]. In the T2DM model, the expression of protein kinase B (Akt, gene name: Akt1) is inhibited, which affects the subsequent phosphorylation process, leading to insufficient Akt activation to carry sufficient signalling forward [45]. In the study, the gene expression of Akt protein in the DC group did not change significantly (Log2 fold change = −0.131), but the expression level was upregulated after intervention using CHPs, which was hypothesised to have possibly been related to the upregulation of 3-phosphoinositide-dependent protein kinase-1 (PDK1/2, gene name: Pdpk1). PDK1/2 can activate Akt protein molecules through phosphorylation [45]. However, insufficient PDK1/2 expression in the DC group may affect further phosphorylation of AKT, thus allowing excessive AKT accumulation, which can be efficiently utilised after PDK1/2 upregulation (CHPs group). Thus, it can be hypothesised that CHPs promote Akt phosphorylation by upregulating the expression of PDK1/2 protein molecules, thereby repairing the insulin-mediated PI3K/Akt signalling pathway. In hepatic glycogen synthesis, phosphorylated Akt promotes the dephosphorylation of glycogen synthase (GYS, gene name: Pygl) by inhibiting the expression of glycogen synthase (GSK3, gene name: Gsk3b) and activating the expression of protein phosphatase (PP1, gene name: Ppp1ca). Glycogen synthase (GYS, gene name: Pygl) promotes the dephosphorylation of hepatic glycogen synthesis [45]. Glucokinase (GCK, gene name: GCK) is the rate-limiting enzyme for insulin-stimulated gluconeogenesis/glycolysis. When the blood glucose concentration in the body exceeds a certain threshold, activated GCK phosphorylates glucose and initiates gluconeogenesis/glycolysis. Furthermore, glucose-6-kinase (G6P, gene name: G6pc3), which also acts as the rate-limiting enzyme for hepatic gluconeogenesis, hydrolyses glucose-6-phosphate to free glucose [46]. Overexpression of phosphoenolpyruvate carboxykinase (PEPCK, gene name: Pck1) increases glucose production and predisposes to hyperinsulinism and insulin resistance [47]. The gene expression level of GCK was upregulated and the expression level of G6pc3 and Pck1 was downregulated in mouse liver by CHPs, contributing to glycolysis and promoting hepatic glycogen synthesis, thus maintaining the homeostasis of glucose metabolism. In the fatty acid synthesis pathway, insulin promotes the production of lipogenic enzymes such as acetyl-CoA carboxylase (ACC, gene name: Acaca) and fatty acid synthase (FAS, gene name: Fas) by upregulating the expression of sterolregulatory element-binding protein-1c (SREBP1c, gene name: Srebf1). Studies have found that high-fructose dietary patterns cause changes in genes such as Acaca and Fas in the liver, leading to lipid accumulation in the liver [48]. After intervention with CHPs, the abnormally altered genes Srebf1, Acaca, and Fas were found to be re-upregulated, thereby promoting normal lipid metabolism.

## 5. Conclusions

In summary, CHP supplementation improved T2DM symptoms in mice and had positive effects on various physiological and biochemical responses. CHPs successfully modulated and corrected metabolic abnormalities such as blood glucose, lipids, and insulin secretion, and had a restorative effect on organ loss caused by a sustained high-fructose diet, slowing the progression of T2DM. This study further demonstrated that CHPs could reverse the aberrantly expressed hepatic genes during T2DM development at the genetic level, thus maintaining the genetic profile of normal mice. We found that CHPs intervened in and alleviated disease progression through multiple glucolipid metabolic pathways, in which the normal functioning of the insulin signalling pathway with IRs-1/PI3K/Akt was maintained. The present study suggests that CHPs, as food-borne components, have the potential for the early prevention and control of T2DM, which may provide a better understanding and theoretical basis for their application in related health foods.

## Figures and Tables

**Figure 1 foods-13-02203-f001:**
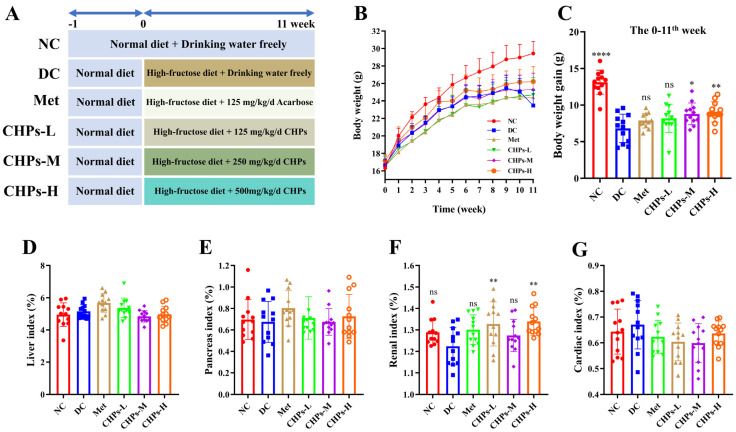
Effect of CHPs on body weight and organ index in T2DM mice (*n* = 12). (**A**) The experimental design of the study. (**B**) Body weight. (**C**) Body weight gain. (**D**) Liver index. (**E**) Pancreas index. (**F**) Renal index. (**G**) Cardiac index. The data are presented as means ± S.D, which were analysed by ANOVA test, followed by Tukey’s test between multiple groups. * indicates significant difference compared with the DC groups, * *p* < 0.05, ** *p* <0.01, **** *p* < 0.0001. ns: not significant.

**Figure 2 foods-13-02203-f002:**
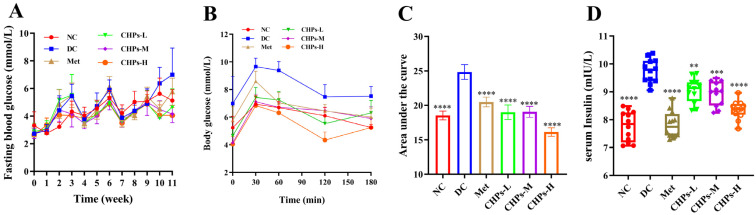
The effects of CHPs on serum glucose related-parameters in T2DM mice. (**A**) Fasting blood glucose. (**B**) OGTT. (**C**) The area under the curve of OGTT. (**D**) Insulin. The data are presented as means ± S.D, which were analysed by ANOVA test, followed by Tukey’s test between multiple groups. * indicates significant difference compared with the DC groups, ** *p* <0.01, *** *p* < 0.001, **** *p* < 0.0001.

**Figure 3 foods-13-02203-f003:**
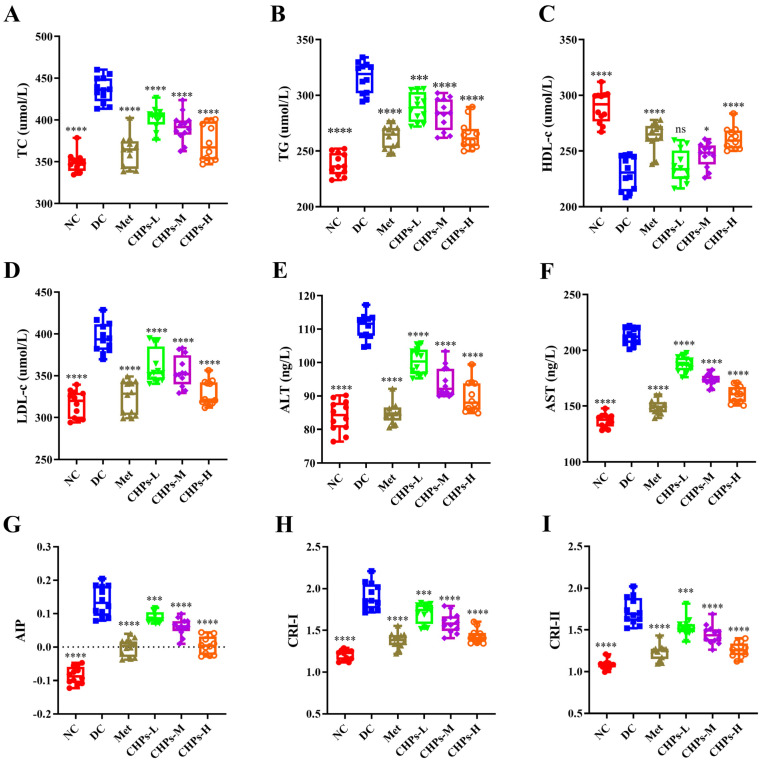
The effects of CHPs on lipid levels and liver function index of serum in T2DM mice (*n* = 12). (**A**) TC, (**B**) TG, (**C**) HDL-c, (**D**) LDL-c, (**E**) ALT, (**F**) AST, (**G**) AIP, (**H**) CRI-I, (**I**) CRI-II. The data are presented as means ± S.D, which were analysed by ANOVA test, followed by Tukey’s test between multiple groups. * indicates significant difference compared with the DC groups, * *p* < 0.05, *** *p* < 0.001, **** *p* < 0.0001. ns: not significant.

**Figure 4 foods-13-02203-f004:**
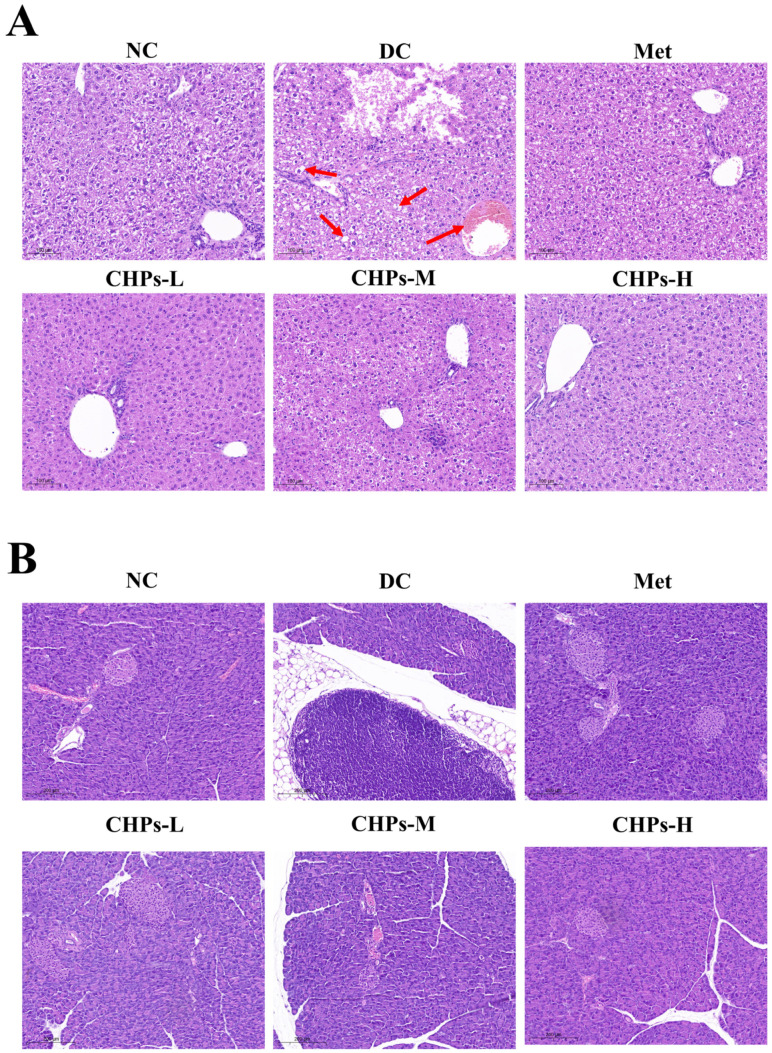
The histopathological images and analyses of the liver and pancreas by HE staining. The image of the liver (**A**) and pancreas (**B**) at 200× magnification. The site indicated by the arrow in the figure represents ballooning from fat accumulation in hepatocytes.

**Figure 5 foods-13-02203-f005:**
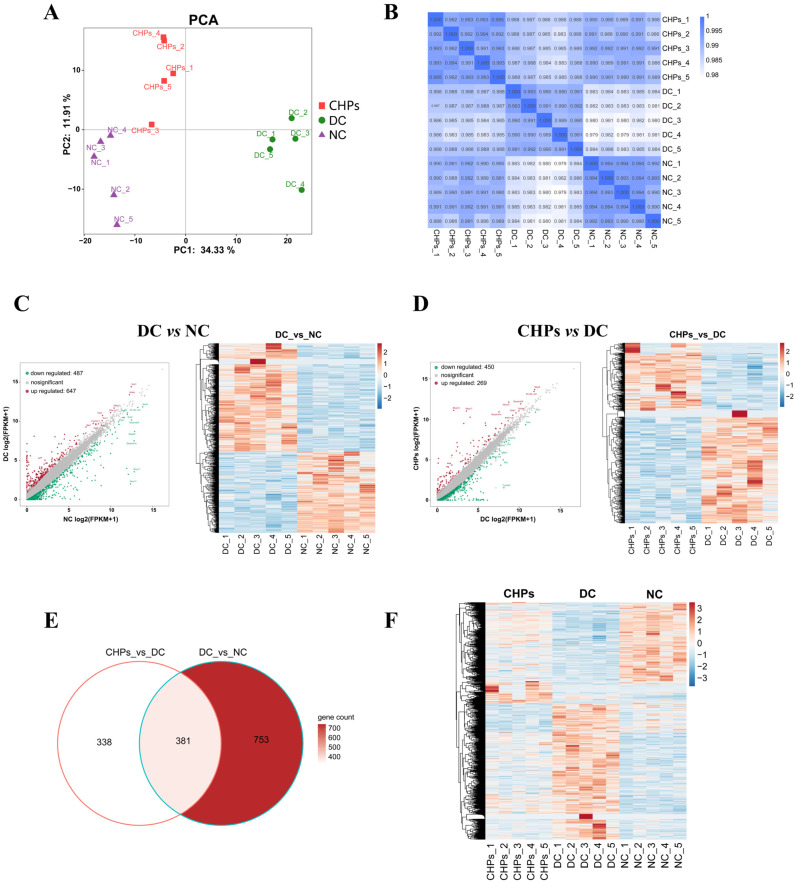
The effect of CHPs on transcriptomics profiles of T2DM mice liver (*n* = 5). (**A**) PCA. (**B**) The correlated heatmap in three groups. (**C**) The number of differentially expressed genes and heatmap of these differentially expressed genes in DC group versus NC group. (**D**) The number of differentially expressed genes and heatmap of these differentially expressed genes in CHP group versus DC group. (**E**) Venn diagrams. (**F**) Hierarchical clustering heatmap of differentially expressed genes in three groups. In the volcano plot analysis, the coloured dots correspond to upregulated (red) and downregulated (green) genes. In the clustering analysis, up- and downregulated genes are coloured red and blue, respectively.

**Figure 6 foods-13-02203-f006:**
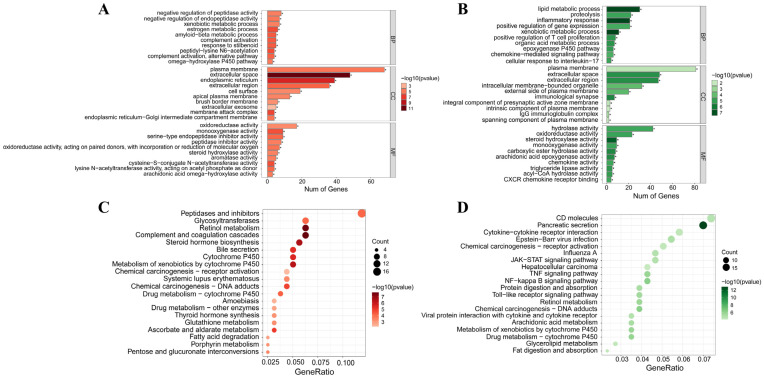
GO (**A**,**B**) analyses and KEGG (**C**,**D**) enrichment of the differentially expressed genes reversed by CHPs in liver tissues. (**A**,**C**) are the results of GO and KEGG enriched by upregulated genes, respectively; (**B**,**D**) are the results of GO and KEGG enriched by downregulated genes, respectively.

**Table 1 foods-13-02203-t001:** Effect of CHPs on insulin sensitivity index in mice (*n* = 12).

Index	NC	DC	Met	CHPs-L	CHPs-M	CHPs-H
HOMA-IR	1.81 ± 0.13 ****	3.03 ± 0.13	2.01 ± 0.12 ****	1.87 ± 0.09 ****	1.65 ± 0.08 ****	1.50 ± 0.06 ****
HOMA-β	89.70 ± 6.24 ****	55.83 ± 2.46	69.05 ± 4.19 **	156.32 ± 7.57 ****	275.12 ± 13.43 ****	303.82 ± 11.99 ****
ISI	−3.70 ± 0.07 ****	−4.22 ± 0.04	−3.81 ± 0.06 ****	−3.74 ± 0.05 ****	−3.61 ± 0.05 ****	−3.52 ± 0.04 ****

The data are presented as means ± S.D, which were analysed by ANOVA test, followed by Tukey’s test between multiple groups. * indicates significant difference compared with the DC groups, ** *p* < 0.01, **** *p* < 0.0001.

**Table 2 foods-13-02203-t002:** Amino acid composition of gastric and GI digestion in sorghum and rice.

Sample	Raw Read	Clean Read	Error Rate (%)	Q_20_/%	Q_30_/%	The Content of GC	Uniquely Mapped Reads	Comparison Ratio/%
NC_1	40,361,870	39,746,718	0.05	98.02	94.18	49	31,954,206	80.39
NC_2	41,166,808	40,378,672	0.05	98.1	94.39	48.04	32,294,451	79.98
NC_3	41,427,642	40,856,278	0.05	97.9	93.79	47.71	33,356,107	81.64
NC_4	40,477,682	39,866,724	0.05	98	94.04	48.26	32,964,838	82.69
NC_5	36,825,102	36,160,176	0.05	98.07	94.19	47.66	29,219,355	80.81
DC_1	43,945,588	43,421,540	0.05	98.26	94.73	49.57	38,602,653	88.9
DC_2	40,764,646	40,313,668	0.05	98.33	94.88	49.46	35,806,617	88.82
DC_3	42,460,782	41,996,324	0.05	98.12	94.34	49.26	36,985,447	88.07
DC_4	41,875,298	41,383,050	0.05	97.77	93.5	49.57	36,529,529	88.27
DC_5	43,072,784	42,493,520	0.05	97.92	93.87	48.66	36,610,646	86.16
CHPs_1	42,978,972	42,386,268	0.05	97.72	93.39	49.53	35,697,806	84.22
CHPs_2	36,564,382	36,134,594	0.05	97.96	93.94	49.08	30,036,435	83.12
CHPs_3	40,873,080	40,404,038	0.05	97.85	93.59	48.71	34,137,945	84.49
CHPs_4	36,089,996	35,695,896	0.05	98.11	94.36	48.78	30,168,364	84.51
CHPs_5	40,848,158	40,239,730	0.05	97.8	93.6	48.56	33,610,245	83.53

Note: Raw read: the number of reads in the raw data, counting the number of sequenced sequences per file in a unit of four lines. Clean read: Number of reads after raw data filtering. Error rate(%): Overall sequencing error rate of data. Q_20_(%); Percentage of total bases with Phred values greater than 20. Q_30_(%): Percentage of total bases with Phred values greater than 30. GC content(%): Percentage of G and C in clean reads for all four bases.

**Table 3 foods-13-02203-t003:** The key gene of insulin signalling pathway in liver tissues.

Gene	Gene ID	DC vs. NC				CHP vs. DC			
		FPKM (DC)	FPKM (NC)	Log2 Fold Change	Regulate	FPKM (CHPs)	FPKM (DC)	Log2 Fold Change	Regulate
Socs1	ENSMUSG00000038037	0.27	0.12	1.052	up	0.20	0.27	−0.378	down
Socs3	ENSMUSG00000053113	18.45	6.03	1.482	up	6.82	18.45	−1.357	down
Irs1	ENSMUSG00000055980	2.72	2.99	−0.263	down	3.60	2.72	0.458	up
Pik3r1	ENSMUSG00000041417	19.10	31.64	−0.859	down	26.19	19.10	0.521	up
Pdpk1	ENSMUSG00000024122	10.19	10.23	−0.131	down	13.24	10.19	0.439	up
Akt1	ENSMUSG00000001729	89.53	63.36	0.369	up	70.03	89.53	−0.295	down
Slc2a4	ENSMUSG00000018566	13.24	3.38	1.852	up	7.70	13.24	−0.742	down
Slc2a2	ENSMUSG00000027690	112.00	100.28	0.023	up	114.50	112.00	0.095	down
Gsk3b	ENSMUSG00000022812	11.16	11.10	−0.118	up	10.19	11.16	−0.076	down
Ppp1ca	ENSMUSG00000040385	140.65	126.29	0.023	up	128.76	140.65	0.066	down
Pygl	ENSMUSG00000021069	309.62	266.45	0.096	up	339.74	309.62	0.189	up
Pygm	ENSMUSG00000032648	0.69	0.33	0.896	up	0.51	0.69	−0.389	down
Gck	ENSMUSG00000041798	98.28	163.35	−0.871	down	132.92	98.28	0.507	up
FBP1	ENSMUSG00000069805	1118.22	751.11	0.449	up	1249.57	1118.22	0.212	down
G6pc3	ENSMUSG00000034793	8.72	5.78	0.464	up	6.12	8.72	−0.449	down
Pck1	ENSMUSG00000027513	720.63	660.65	0.004	up	497.70	720.63	−0.561	down
Srebf1	ENSMUSG00000020538	326.54	295.36	0.013	up	270.46	326.54	−0.211	down
Acaca	ENSMUSG00000020532	287.22	66.27	1.998	up	138.54	287.22	−1.005	down
Fasn	ENSMUSG00000025153	1741.45	515.35	1.640	up	644.10	1741.45	−1.389	down

## Data Availability

The original contributions presented in the study are included in the article, further inquiries can be directed to the corresponding author.

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
