# Peer review of "Anti-Type II Diabetic Effects of Coix Seed Prolamin Hydrolysates: Physiological and Transcriptomic Analyses"

_foods, 2024, doi:10.3390/foods13142203_

Round 1

Reviewer 1 Report

Comments and Suggestions for Authors

The article (foods-3052369) entitled “Anti-type II diabetic effects of coix seed prolamin hydrolysates: physiological and transcriptomic analyses” aims to evaluate the preventive, therapeutic, and modifying effects of CHPs on T2DM by combining physiological and liver transcriptomics with a T2DM mouse model. Intervention with CPHs effectively prevented weight loss; regulated abnormal changes in blood glucose; inhibited abnormal blood lipid levels; alleviated insulin resistance; and restored pancreatic islet tissue function in mice fed a high-fructose diet. In addition, CHPs also protect various organ tissues and are effective in preventing damage to the liver and pancreatic islet cells. It is also found that CHPs reversed the abnormally altered hepatic gene profile in model mice. Also, CHPs play a positive role in insulin signaling pathway dominated by the IRS-1/PI3K/AKT pathway. Thus, CHPs can be used as effective food-borne glucose-modifying components of healthy foods.

The study adds valuable results and is well written.

In line 28, metabolic pathways. we demonstrated                        

Please change to metabolic pathways. W demonstrated

All abbreviations must be written in full.

For example, IRS-1/PI3K/AKT

Please give more details about the insulin signaling pathway dominated by the IRS-1/PI3K/AKT pathway.

Several figures need to be presented in a better and clearer way.

For example, figures 5 and 6.

Please compare your study with other recent studies.

Some references are not complete.  

Please add years. For example:

  1. Naseri, K.; Saadati, S.; Sadeghi, A.; Asbaghi, O.; Ghaemi, F.; Zafarani, F.; Li, H.B.; Gan, R.Y. The Efficacy of Ginseng (Panax) on Human Prediabetes and Type 2 Diabetes Mellitus: A Systematic Review and Meta-Analysis. Nutrients. 14(12), 2401.  https://doi.org/10.3390/nu14122401.
  2. Tan, Y.; Tan, S.; Ren, T.; Yu, L.; Li, P.; Xie, G.; Chen, C.; Yuan, M.; Xu, Q.; Chen, Z. Transcriptomics Reveals the Mechanism of 657 Rosa roxburghii Tratt Ellagitannin in Improving Hepatic Lipid Metabolism Disorder in db/db Mice. Nutrients. 15(19), 4187. 658 https://doi.org/.
  3. Huang, Z.; Liu, Y.; Liu, X.; Chen, K.; Xiong, W.; Qiu, Y.; He, X.; Liu, B.; Zeng, F. Sanghuangporus vaninii mixture ameliorated 660 type 2 diabetes mellitus and altered intestinal microbiota in mice. Food Funct. 13(22), 11758–11769. 661 https://doi.org/10.3390/nu15194187.

Comments on the Quality of English Language

The article is generally well written.

Author Response

Dear Experts:

Thank you for your letter and for the reviewers’ comments concerning our manuscript entitled Anti-Type II Diabetic Effects of Coix Seed Prolamin Hydrolysates: Physiological and Transcriptomic Analyses (ID: foods-3052369). Those comments are all valuable and very helpful for revising and improving our paper, as well as the important guiding significance to our researches. We have studied comments carefully and have made correction which we hope meet with approval. 

Improved portion are marked in red in the revised manuscripts. Because the most current version has added some words, this caused the line numbers has been changed. The main corrections in the paper and the responds to the editor’s and review’s comments are in following next page. Expect the editor to allow the request.

If you require any additional information regarding our manuscript, please do not hesitate to contact us directly via the resources below. Thank you for your time and consideration.

Thank you and best regards.

Sincerely,

Zhi-Ming Li

Heilongjiang Bayi Agricultural University, 163319, Daqing, China

E-mail: lizhiming1998@126.com

Responds to the reviewer’s comments:

Reviewer #1:

  • Comment 1:In line 28, metabolic pathways. we demonstrated. Please change to metabolic pathways. W demonstrated.

Answer: Thank you for your suggestion. We have corrected the two errors you pointed out. Please refer to line 28 of the manuscript.

  • Comment 2:All abbreviations must be written in full. For example, IRS-1/PI3K/AKT.

Answer: Thank you for your suggestion. The error here is to display the full name of the abbreviated name when it first appears. We have already added to this, please refer to the 30-31 lines. At the same time, we examined the manuscript and made changes where there were similar problems.

  • Comment 3:Please give more details about the insulin signaling pathway dominated by the IRS-1/PI3K/AKT pathway.

Answer: Thank you for your suggestion. We added more details about the IRS-1/PI3K/AKT pathway in lines 475-495 of the manuscript and marked them in red. The details are as follows.

Insulin signaling is triggered by the binding of insulin to the insulin receptor on the target cell membrane. Insulin receptor substrates that have been activated activate downstream signaling pathways to perform the biological functions of insulin. Among them, the PI3K-Akt pathway is one of the important pathways for insulin to fulfill its biological functions [1]. The mediating pathways can be roughly summarized as follows [2]: when extracellular insulin signaling is delivered to the cell via IR and IRS, the SH2 domain of the regulatory subunit of PI3K, p85, will bind to the phosphorylated IRS, thereby activating the catalytic subunit of PI3K, p110. In response to growth hormone stimulation, the p110 subunit catalyzes phosphatidylinositol (4,5)-bisphosphate (PIP2) into phosphoinositide (3,4,5)-triphosphate (PIP3), and PIP3 ultimately activates AKT and other downstream factors. PIP3 binds to the PH structural domain of AKT, thereby promoting AKT phosphorylation by upstream kinases. First, 3-phosphoinositide-dependent protein kinase-1, (PDK1) phosphorylates the catalytic structural domain of AKT, Thr308, and thus 10% of AKT activity is activated. Then, the mammalian target of rapamycin complex 2 (mTORC2), DNA-dependent protein kinase (DNA-PK), and ataxia telangiectasia mutated kinase (ATM) phosphorylate the AKT regulatory domain Ser473, thereby fully activating the kinase activity of AKT. Activated AKT is released from the plasma membrane and translocates to the cytoplasm, mitochondria, and nucleus to phosphorylate various substrates. Important substrates of AKT are glycogen synthase kinase 3β (GSK3β), forkhead box protein O1 (FOXO1) and protein kinase B substrate 160 (AKT substrate 160, AS160), which regulate glycogen synthesis, gluconeogenesis and glucose uptake, respectively.

[1] Mackenzie, R. W.; Elliott, B. T. Akt/PKB activation and insulin signaling: a novel insulin signaling pathway in the treatment of type 2 diabetes. Diabetes, metabolic syndrome and obesity: targets and therapy, 2014, 7, 55-64. https://doi.org/10.2147/DMSO.S48260.

[2] Wu, F.; Shao, Q.Q.; Xia, Q.S.; Hu, M.L.; Zhao, Y.; Wang, D.K; Fang, K.; Xu, L.J.; Zou, X.; Chen, Z.; Chen, G.; Lu, F.E. A bioinformatics and transcriptomics based investigation reveals an inhibitory role of Huangli-an-Renshen-Decoction on hepatic glucose production of T2DM mice via PI3K/Akt/FoxO1 signaling pathway. Phytomedicine, 2021, 83, 153487. https://doi.org/ 10.1016/j.phymed.2021.153487.

  • Comment 4:Several figures need to be presented in a better and clearer way. For example, figures 5 and 6.

Answer: Thank you for your suggestion. We have adjusted the size and presentation of Figures 5 and 6.

  • Comment 5:Please compare your study with other recent studies.

Answer: Thank you for your suggestion. We have compared other similar studies reported in lines 515-528 of the manuscript and marked them in red. The details are as follows.

Similar research involving protein hydrolysate interventions in T2DM have yielded significant results, such as Wang et al. who fed walnut protein hydrolysate (3-10 kDa) at concentrations of 200, 500, and 800 mg/kg/d orally to T2DM mice (n=10), and reported that after 4 weeks, the mice showed a 64.82% reduction in fasting glucose levels and a 23.71% increase in insulin secretion [1] . Yan et al. gavaged ICR mice (n=6) with wheat germ protein hydrolysate (>5 kDa, 3.24 mg/mL) at a dose of 48.6 mg/(kgd), and this. Led to alleviation of polydipsia, polyphagia, and hyperglycaemia after 2 weeks, with effective control of postprandial glycaemia [2]. In these cases, it can be found that most of the food-borne peptides used for in vivo experiments are mixed peptide. Most of these hydrolysates are obtained through simple treatments such as desalination and decolorization. This actually fits the application scenario of food-borne peptides in dietary nutrition and health interventions and has the advantage of low costs. In the future, this product form may be the direction of research and commercialization of coix seed prolamin peptides.

[1] Wang, J.; Du, K.Y.; Fang, L.; Min, W.H.; Liu, J.S. Evaluation of the antidiabetic activity of hydrolyzed peptides derived from Juglans mandshurica Maxim. fruits in insulin‐resistant HepG2 cells and type 2 diabetic mice. J. Food Biochem. 2018, 42(3), e12518. https://doi.org/10.1111/jfbc.12518.

[2] Yan, H.; Zhang, Q.; Jiang, M.Z.; Zhu, S.H.; Nie, X.D.; Yu, Y.J.; Zhang, J.X.; Jia, J.Q.; Xiong, M. Isolation and structural identification of hypoglycemic peptides from wheat germ protein. Food Sci. 2018, 39(20), 92-98. https://doi.org/10.7506/spkx1002-6630-201820014.

  • Comment 6:Some references are not complete. Please add years. For example:
  1. Naseri, K.; Saadati, S.; Sadeghi, A.; Asbaghi, O.; Ghaemi, F.; Zafarani, F.; Li, H.B.; Gan, R.Y. The Efficacy of Ginseng (Panax) on Human Prediabetes and Type 2 Diabetes Mellitus: A Systematic Review and Meta-Analysis. Nutrients. 14(12), 2401.  https://doi.org/10.3390/nu14122401.
  1. Tan, Y.; Tan, S.; Ren, T.; Yu, L.; Li, P.; Xie, G.; Chen, C.; Yuan, M.; Xu, Q.; Chen, Z. Transcriptomics Reveals the Mechanism of 657 Rosa roxburghii Tratt Ellagitannin in Improving Hepatic Lipid Metabolism Disorder in db/db Mice. Nutrients. 15(19), 4187. 658 https://doi.org/.
  1. Huang, Z.; Liu, Y.; Liu, X.; Chen, K.; Xiong, W.; Qiu, Y.; He, X.; Liu, B.; Zeng, F. Sanghuangporus vaninii mixture ameliorated 660 type 2 diabetes mellitus and altered intestinal microbiota in mice. Food Funct. 13(22), 11758–11769. 661 https://doi.org/10.3390/nu15194187.

Answer: Thank you for your suggestion. We have added "year" information to references 1, 19 and 20. Please refer to lines 643, 704 and 707 of the manuscript respectively. Additional information has been marked in red.

Reviewer 2 Report

Comments and Suggestions for Authors

The article is interesting, the novelty is clearly described. Nevertheless, the rodent model is confusing, authors assessed an antidiabetic effect; nevertheless, the diet-induced model is an obese phenotype with diverse metabolic alterations, like insulin resistance, dislypidemia, and hepatic steatosis. Authors must clarify this major concern prior further evaluation.

Author Response

Dear Experts:

Thank you for your letter and for the reviewers’ comments concerning our manuscript entitled Anti-Type II Diabetic Effects of Coix Seed Prolamin Hydrolysates: Physiological and Transcriptomic Analyses (ID: foods-3052369). Those comments are all valuable and very helpful for revising and improving our paper, as well as the important guiding significance to our researches. We have studied comments carefully and have made correction which we hope meet with approval. 

Improved portion are marked in red in the revised manuscripts. Because the most current version has added some words, this caused the line numbers has been changed. The main corrections in the paper and the responds to the editor’s and review’s comments are in following next page. Expect the editor to allow the request.

If you require any additional information regarding our manuscript, please do not hesitate to contact us directly via the resources below. Thank you for your time and consideration.

Thank you and best regards.

Sincerely,

Zhi-Ming Li

Heilongjiang Bayi Agricultural University, 163319, Daqing, China

E-mail: lizhiming1998@126.com

Responds to the reviewer’s comments:

Reviewer #2:

  • Comment 1:The article is interesting, the novelty is clearly described. Nevertheless, the rodent model is confusing, authors assessed an antidiabetic effect; nevertheless, the diet-induced model is an obese phenotype with diverse metabolic alterations, like insulin resistance, dislypidemia, and hepatic steatosis. Authors must clarify this major concern prior further evaluation.

Answer: Thank you for your comments. A high-fructose diet was used in this study to induce T2DM in mice with disturbed glucolipid metabolism. We are mainly motivated by two considerations. First, from a practical point of view, the high intake of high fructose in daily life, which is a causative factor of various metabolic diseases, including type II diabetes, hypertension, hyperuricemia, and non-alcoholic fatty liver disease, etc., has not yet required sufficient attention in intervention studies in this area. The second is that the animal model created by the high fructose diet has symptoms of glucose-lipid metabolism disorders such as insulin resistance, impaired pancreatic islet function, hyperglycemia, and hyperlipidemia, which can satisfy the research objectives of this experiment. Therefore, we have made additions to the reasons and scientific significance of using the high fructose-induced T2DM model. The additions are shown in lines 83-93 of the manuscript and have been labeled in red font. The details are as follows.

With the improvement of living standards, fructose is widely used as a sweetener in the food industry, which leads to a large amount of intake because of its low price, good taste and low satiety. High fructose intake induces insulin resistance. Metabolism of fructose in the liver leads to high lipid production, elevated free fatty acids and impaired insulin signaling receptors, which reduces tissue response to insulin signaling [1]. In addition, fat from fructose metabolism accumulates in the liver, causing non-alcoholic fatty liver disease, which affects the normal function of the liver and ultimately produces insulin resistance, in which hepatic fructokinase plays a key role in the development of insulin resistance [2][3]. Therefore, it is scientifically and practically important to examine the regulatory role of CHPs on T2DM (typically characterized by dysglycemia and insulin resistance) induced by high-fructose diets.

[1]Nakagawa, T.; Johnson, R.J.; Andre-Hernando, A.; Roncal-Jimenez, C.; Sanchez-Lozada, L.G.; Tolan, D.R.; Lanaspa, M.A. Fructose production and metabolism in the kidney. J. Am. Soc. Nephrol. 2020, 31(5), 898-906. https://doi.org/10.1681/asn.2019101015.

[2]Jarukamjorn, K.; Chatuphonprasert, W.; Jearapong, N.; Punvittayagul, C.; Wongpoomchai, R. Tetrahydrocurcumin attenuates phase I metabolizing enzyme-triggered oxidative stress in mice fed a high-fat and high-fructose diet. J. Funct. Foods. 2019, 55, 117-125. https://doi.org/10.1016/j.jff.2019.02.021.

[3]Andres-Hernando, A.; Orlicky, D. J.; Kuwabara, M.; Ishimoto, T.; Nakagawa, T.; Johnson, R. J.; Lanaspa, M. A. Deletion of fructokinase in the liver or in the intestine reveals differential effects on sugar-induced metabolic dysfunction. Cell metab. 2020 32(1), 117-127. https://doi.org/10.1016/j.cmet.2020.05.012.

Reviewer 3 Report

Comments and Suggestions for Authors

In the manuscript ‘Anti-Type II Diabetic Effects of Coix Seed Prolamin Hydrolysates: Physiological and Transcriptomic Analyses’ evaluated the preventive, therapeutic, and modifying effects of CHPs on T2DM by combining physiological and liver transcriptomics with a T2DM mouse model. Although the manuscript is well written, neat, and the material and methods well explained, in the reviewer's opinion, the manuscript has criticism.

Comments:

-          The English throughout the article must be widely checked. Countless grammatical and typographical errors (commas and spaces). The article must be checked in depth.

-          Numerous phrases require a verb. examples on lines 66 and 140.

-          Line 56: Is protein the main component of coix seed? and carbohydrates?

-          The authors should discuss the weight loss caused by diabetes in rats (as opposed to normal diet), and shown similar results in other articles.

-          How was the adipose tissue in these rats? The authors observe a decrease in weight in the DC group, but then they have elevated TC and TG levels.

-          Redundant final objective

-          Line 125: what do you mean by the samples?

-          The material and methods of lines 125-130 should be better explained.

-          line 132. different reference

-          Line 224: slower wirht fain in mice from othre groups--> Specify which groups.

-          Lines 236-238. increase in the liver and heart indices and a decline in the kidney and pancreas indices.  The reviewer does not identify where these increases and decreases are.

-          I do not identify the need to add renal index and cardiac index in the article. Explain and discuss.

-          Figure 1A: DC--> normal diet + drinking water freeely. What difference is there then with respect to NC?

-          Figure legend (C): bodyweignt gain needs a space.

-          Figure 2: Missing statistics in Figures A and B.

-          Have the authors thought about the atherogenic indices AIP, CRI and CRII?

-          How do the authors differentiate ballooning from fat accumulation in hepatocytes ? It would be interesting to show ballooning with an arrow.  The same with cellular infiltration.

-          Have the authors thought of performing the NAFLD score?

-          During section 3 results are discussed. In fact the title of the section is 3. results and discussion. But then we find a section 4. discussion. It is important to separate or merge them.

- Some figures could be made larger for a better reading of the article.

Author Response

Dear Experts:

Thank you for your letter and for the reviewers’ comments concerning our manuscript entitled Anti-Type II Diabetic Effects of Coix Seed Prolamin Hydrolysates: Physiological and Transcriptomic Analyses” (ID: foods-3052369). Those comments are all valuable and very helpful for revising and improving our paper, as well as the important guiding significance to our researches. We have studied comments carefully and have made correction which we hope meet with approval. 

Improved portion are marked in red in the revised manuscripts. Because the most current version has added some words, this caused the line numbers has been changed. The main corrections in the paper and the responds to the editor’s and review’s comments are in following next page. Expect the editor to allow the request.

If you require any additional information regarding our manuscript, please do not hesitate to contact us directly via the resources below. Thank you for your time and consideration.

Thank you and best regards.

Sincerely,

Zhi-Ming Li

Heilongjiang Bayi Agricultural University, 163319, Daqing, China

E-mail: lizhiming1998@126.com

Responds to the reviewer’s comments:

Reviewer #3:

  • Comment 1:The English throughout the article must be widely checked. Countless grammatical and typographical errors (commas and spaces). The article must be checked in depth.

Answer: Thank you for your comments. We have carried out professional English polishing on Editage (www.editage.cn) platform according to your suggestion. The supporting materials are shown in the figure below. At the same time, we have rechecked and revised the typographical issues in the manuscript, please refer to the manuscript for the details.

  • Comment 2:Numerous phrases require a verb. examples on lines 66 and 140.

Answer: Thank you for your comments. We have corrected these two grammatical errors; please consult lines 67-68 and 152 of the manuscript, respectively. In addition, we have rechecked the manuscript for similar grammatical problems and corrected them, please refer to the manuscript for the details.

  • Comment 3:Line 56: Is protein the main component of coix seed? and carbohydrates?

Answer: Thank you for your comments. According to the literature, the chemical component of coix seed include the following: crude starch (~70%)[1], protein (16%~17%)[2], lipids (5.1%~9.4%)[3], total carotenoid (10 mg β-carotene equivalent per 100 g of grain, wet basis)[4], the total phytosterol (4733 mg/kg)[5]. The references are shown below. This shows that starch and protein are the two main chemical components of coix seed. In addition, we have modified this sentence in order to more accurately describe the main ingredient in coix seed, please refer to lines 55-58 in the manuscript for details.

[1] Corke, H., Huang, Y., & Li, J. S. (2016). Coix: Overview. In C. Wrigley, H. Corke, K. Seetharaman, & J. Faubion (Eds.), Encyclopedia of Food Grains (pp. 184e189). Oxford: Academic Press.

[2] Huda, M. N., Li, X., Jahan, T., He, Y., Guan, C., Zhang, K., ... & Zhou, M. (2023). Acceleration of the genetic gain for nutraceutical improvement of adlay (Coix L.) through genomic approaches: Current status and future prospects. Food Reviews International, 39(8), 5377-5401.

[3] Liu, X., Zhang, X., Rong, Y. Z., Wu, J. H., Yang, Y. J., et al. (2015a). Rapid determination

of fat, protein and amino acid content in coix seed using near-infrared spectroscopy technique. Food Analytical Methods, 8, 334e342.

[4] Choi, Y., Jeong, H. S., & Lee, J. (2007). Antioxidant activity of methanolic extracts from some grains consumed in Korea. Food Chemistry, 103, 130e138.

[5] Wu, T. T., Charles, A. L., & Huang, T. C. (2007). Determination of the contents of the main biochemical compounds of adlay (Coxi lachrymal-jobi). Food Chemistry, 104, 1509e1515.

  • Comment 4:The authors should discuss the weight loss caused by diabetes in rats (as opposed to normal diet), and shown similar results in other article

Answer: Thank you for your suggestion. Our results show that a high fructose diet delays normal development and limits body weight gain in mice. However, this symptom was reduced with the intervention of CPHs, which inhibited wasting and restored the ability of mice to gain normal weight. Previous studies have reported that excessive catabolism and utilisation of body fat and proteins in mice with progressive disease can lead to impaired weight gain and weight loss due to insufficient glucose utilisation. At the suggestion of the reviewers, we have described the reasons for this phenomenon in the "Discussion" section, which can be found in lines 502-504 of the manuscript. In addition, we compared the results of this study with other studies, and the modifications can be seen in lines 242-243 of the manuscript.

  • Comment 5:How was the adipose tissue in these rats? The authors observe a decrease in weight in the DC group, but then they have elevated TC and TG levels.

Answer: Thank you for your comments. Sorry, we did not consider physiological and biochemical analyses of adipose tissue in mice in our experimental design at that time. Considering the further evaluation of the hypoglycemic efficacy of coleus coix seed prolamin peptides, this issue will be considered and designed by our group in the future follow-up studies. Thank you again for raising this issue! The results of the study showed that the mice lost weight after high fructose dietary treatment.This could be because excessive catabolism and utilisation of body fat and proteins in mice with progressive disease can lead to impaired weight gain and weight loss due to insufficient glucose utilisation. At the same time, both TG and TC were increased in the mice, which indicated symptoms of lipid metabolism disorders. The coexistence of the two phenomena of weight loss and increased TG and TC during the formation of the T2DM mouse model has been similarly reported in other studies. For example, studies by Zhu et al.[1] and Li et al.[2][3] Reported.

[1] Zhu, X.A.; Qiu, Z.R.; Ouyang, W.; Miao, J.Y; Xiong, P.; Mao, D.B; Feng, K.L.; Li, M.X.; Luo, M.N.; Xiao, H.; Cao, Y. Hepatic transcriptome and proteome analyses provide new insights into the regulator mechanism of dietary avicularin in diabetic mice. Food Res. Int. 2019, 125, 108570.

[2] Li, Y.; Chen, D.; Xu, C.; Zhao, Q.; Ma, Y.; Zhao, S.; Chen, C. Glycolipid metabolism and liver transcriptomic analysis of the therapeutic effects of pressed degreased walnut meal extracts on type 2 diabetes mellitus rats. Food Funct. 2020, 11(6), 5538-5552.

[3] Li, Y.L.; Chen, D.; Zhang, F.; Li, Y.P.; Ma, Y.G.; Zhao, S.L.; Chen, C.Y.; Wang, X.S.; Liu, J. Preventive effect of pressed degreased walnut meal extracts on T2DM rats by regulating glucolipid metabolism and modulating gut bacteria flora. J. Funct. Foods. 2020, 64, 103694.

  • Comment 6:Redundant final objective

Answer: Thank you for your suggestion. We guess that you are referring to lines 85-87 of the "Introduction" in the original manuscript: This study ultimately aim to provide supporting data for the use of coix seed prolamin peptides in the prevention and treatment of T2DM. We've removed this sentence based on your suggestion.

  • Comment 7:Line 125: what do you mean by the samples?

Answer: Thank you for your comments."Sample" refers to the different amounts of CHPs added in each treatment group. For clarity and accuracy, we have modified this sentence to read "Different amounts of CHPs were dissolved in saline by gavage (2 mL) for 11 weeks, and the specific experimental design is shown in Fig. 1A.". The revised sentence is on lines 136-137 of the manuscript and has been marked in red font.

  • Comment 8:The material and methods of lines 125-130 should be better explained.

Answer: Thank you for your suggestion. We have revised this section in accordance with your comments to ensure the accuracy of the material and methods. Specific details of the revisions can be found in lines 136-143 of the manuscript, which have been highlighted in red font. The details are as follows.

Different amounts of CHPs were dissolved in saline by gavage (2 mL) for 11 weeks, and the specific experimental design is shown in Fig. 1A. Blood samples of each group were collected from tail vein, and the blood glucose levels were measured using a glucometer according to the manufacturer’s instructions (Sannuo Biosensor Co., Ltd, Hunan, China).After 10 and 11 weeks of intervention, the blood glucose levels of mice in the DC group were observed to be greater than 6.1 mmol/L for two consecutive week. This meets the clinical diagnostic criteria of "prodiabetes".

  • Comment 9:line 132. different reference.

Answer: We are very sorry and thank you very much for catching our mistake. The "2" was a writing error. We've changed "2" to "[2]". Thank you again for your comments.

  • Comment 10:Line 224: slower wirht fain in mice from othre groups--> Specify which groups.

Answer: Thank you for your comments. Sorry, this is an error in description. The point being made here is that mice fed high fructose gained weight at a lower rate than the NC group, including the DC, Met, CHPs-L, CHPs-M, and CHPs-H groups. We have modified this sentence to ensure that it more accurately expresses the results of the trial. For details see lines 235-236 in the manuscript.

  • Comment 11:Lines 236-238. increase in the liver and heart indices and a decline in the kidney and pancreas indices.  The reviewer does not identify where these increases and decreases are.

Answer: Thank you for your comments. As shown in Table 1, we tabulated the results of the organ index test. Compared with the NC group, the DC group showed an increase in the liver and heart indices, and a decline in the kidney and pancreas indices. In addition, the phrase "with the decline in the kidney index reaching a significant difference (p<0.05)" in this sentence was an inappropriate analysis, and we have removed this phrase. Please consult lines 248-249 of the manuscript. Thank you again for your comments.

Table 1 Effect of CHPs on body weight and organ index in T2DM mice (n=12)

group

Liver index (%)

Pancreas index (%)

Renal index (%)

Cardiac index (%)

NC

4.95±0.73

0.70±0.19

1.29±0.06

0.64±0.09

DC

5.15±0.41

0.67±0.09

1.22±0.19

0.67±0.09

Met

5.68±0.57

0.80±0.17

1.33±0.07

0.60±0.06

CHPs-L

5.37±0.60

0.71±0.20

1.30±0.10**

0.62±0.07

CHPs-M

4.84±0.35

0.67±0.12

1.27±0.08

0.60±0.11

CHPs-H

4.97±0.44

0.73±0.20

1.34±0.06**

0.63±0.05

The data were presented as means ± S.D, which were analyzed by ANOVA test, followed by Tukey’s test between multiple groups. * indicates significant difference compared with the DC groups, **p <0.01

  • Comment 12:I do not identify the need to add renal index and cardiac index in the article. Explain and discuss.

Answer: Thank you for your comments. The two metrics, renal index and cardiac index, were shown in the article for research purposes. High fructose intake has been reported to lead to the development of metabolic syndromes such as hyperuricemia and T2DM. On the one hand, fructose decreases renal excretion of uric acid, which leads to an increase in blood uric acid by a mechanism related to insulin resistance [1]. On the other hand, fructose-induced elevation of blood glucose and uric acid will also make the burden on the kidneys heavier, stimulate the kidneys to produce inflammatory responses and oxidative stress, causing damage to the kidneys [2]. Numerous previous studies have reported that an increase in the lipid level is an important risk factor for the development of cardiovascular disease [3]. The results of the present study suggest that mice on a high-fructose diet show signs of lipid metabolism disorders, which may lead to cardiovascular disease. In summary, two physiological indices, renal index and cardiac index, were measured in this study.

[1] Jamnik, J.; Rehman, S.; Mejia, S. B.; de Souza, R. J.; Khan, T. A.; Leiter, L. A.; Wolever, T. M.; Kendall, C. W.; Jenkins, D. J.; Sievenpiper, J. L. Fructose intake and risk of gout and hyperuricemia: a systematic review and meta-analysis of prospective cohort studies. Nutr. Metab. 2016, 6(10), e013191.

[2] Lu, Y.; Wu, Y.; Chen, X.; Yang, X.; Xiao, H. Water extract of shepherd's purse prevents high-fructose induced-liver injury by regulating glucolipid metabolism and gut microbiota. Food Chem. 2021, 342, 128536.

[3] Sharif, S.; Van Der Graaf, Y.; Nathoe, H. M.; de Valk, H. W.; Visseren, F. L.; Westerink, J.; SMART Study Group. Esteghamati, Comment on Sharif et al. HDL Cholesterol as a residual risk factor for vascular events and all-cause mortality in patients with type 2 diabetes. Diabetes Care. 2016, 39, e189.

  • Comment 13:Figure 1A: DC--> normal diet + drinking water freeely. What difference is there then with respect to NC?

Answer: We are very sorry and thank you very much for catching our mistake. We have modified Figure 1A by changing "DC--> normal diet + drinking water freeely" to "DC--> High-fructose diet + drinking water freeely".

  • Comment 14:Figure legend (C): bodyweignt gain needs a space.

Answer: We are very sorry and thank you very much for catching our mistake. We have changed "bodyweignt gain" to "body weignt gain" in line 258 of the manuscript.

  • Comment 15:Figure 2: Missing statistics in Figures A and B.

Answer: Thank you for your comments. Figure 2A was designed to show the changes in blood glucose levels in each group of mice during the intervention. Figure 2B is to show the fluctuation of blood glucose levels in each group during the OGTT test. The authors designed the graph in this way in order to be able to observe the blood glucose levels of the mice in a simple and intuitive way. As reported in the literature, the presentation of these two indicators is consistent with the way data are counted in conventional studies. The relevant literature is as follows.

Wen, J. J., Li, M. Z., Hu, J. L., Wang, J., Wang, Z. Q., Chen, C. H., Nie, S. P. (2023). Different dietary fibers unequally remodel gut microbiota and charge up anti-obesity effects. Food Hydrocolloids, 140, 108617.

Li, Y., Chen, D., Xu, C., Zhao, Q., Ma, Y., Zhao, S., & Chen, C. (2020). Glycolipid metabolism and liver transcriptomic analysis of the therapeutic effects of pressed degreased walnut meal extracts on type 2 diabetes mellitus rats. Food & function, 11(6), 5538-5552.

  • Comment 16:Have the authors thought about the atherogenic indices AIP, CRI and CRII?

Answer: Thank you for your suggestion. We are very sorry that our previous experimental design did not take these three biochemical indicators into account. Indeed, further examination of AIP, CRI and CRII in the present study could help to evaluate lipid metabolism in mice more comprehensively, and at the same time could reflect the preventive effect of CHPs on related cardiovascular diseases. Due to the limitations of the experimental period, relevant the atherogenic indices, such as AIP, CRI and CRII, will be designed in our future experiments to further examine the effect of CHPs treatment on lipid metabolism. Thanks again to the reviewers for this helpful suggestion!

  • Comment 17:How do the authors differentiate ballooning from fat accumulation in hepatocytes ? It would be interesting to show ballooning with an arrow.  The same with cellular infiltration.

Answer: Thank you for your suggestion. This does make it clearer and more straightforward. We have revised Figure 4 as requested by the reviewer, please refer to Figure 4 of the manuscript for details of the revision.

  • Comment 18:Have the authors thought of performing the NAFLD score?

Answer: Thank you for your suggestion. It has been reported that sustained intake of high fructose induces NAFLD. On the one hand, the fat produced by fructose metabolism accumulates in the liver, which can cause non-alcoholic fatty liver disease and affect the normal function of the liver [1]. On the other hand, high fructose intake significantly increases uric acid production, and elevated blood uric acid leads to an increased risk of NAFLD development [2][3]. By assessing the NAFLD score contributes to a more comprehensive assessment of the ameliorative effects of CHPs on liver injury. However, for the purpose of this study, this paper is focusing on the modulatory effects of CHPs on high fructose intake-induced disorders of glucose-lipid metabolism. Focused on evaluating the effects of CHPs on glucose and lipid metabolism in mice, and therefore failed to use the NAFLD score as a primary metric. Considering the comprehensive evaluation of the efficacy of CHPs for the modulation of metabolic syndrome, we will focus on this component in future trial designs. Thanks again to the reviewers for this helpful suggestion!

[1] Jarukamjorn, K.; Chatuphonprasert, W.; Jearapong, N.; Punvittayagul, C.; Wongpoomchai, R. Tetrahydrocurcumin attenuates phase I metabolizing enzyme-triggered oxidative stress in mice fed a high-fat and high-fructose diet. J. Funct. Foods. 2019, 55, 117-125.

[2] Sayehmiri, K.; Ahmadi, I.; Anvari, E. Fructose feeding and hyperuricemia: A systematic review and meta-analysis. Clin. Nutr. Res. 2020, 9(2), 122.

[3] Xu, K.; Zhao, X.; Fu, X.; Xu, K.; Li, Z.; Miao, L.; Li, Y.; Cai, Z.; Qiao, L.; Bao, J. Gender effect of hyperuricemia on the development of nonalcoholic fatty liver disease (NAFLD): A clinical analysis and mechanistic study. Biomed. Pharmacother. 2019, 117, 109158.

  • Comment 19:During section 3 results are discussed. In fact the title of the section is 3. results and discussion. But then we find a section 4. discussion. It is important to separate or merge them.

Answer: Thank you for your suggestion. We have changed "3 Results and Discussion" to "Results" in line 230 of the manuscript and marked it in red.

  • Comment 20:Some figures could be made larger for a better reading of the article.

Answer: Thank you for your suggestion. We have modified the dimensions of the images in the manuscript as suggested by the reviewers.

Round 2

Reviewer 3 Report

Comments and Suggestions for Authors

1. Table 1: during the text, it must be reflected that there are no significant differences in liver, pancreas and cardiac indices

2. I do not understand why the authors say that they cannot make the CRI, CRII and AIP indices if they have the individual values ​​of TC, HDL, LDL and TG.

3. Figure 4: indicate in the legend what the red arrow points to.

Author Response

Dear Experts:

Thank you for your letter and for the reviewers’ comments concerning our manuscript entitled Anti-Type II Diabetic Effects of Coix Seed Prolamin Hydrolysates: Physiological and Transcriptomic Analyses (ID: foods-3052369). Those comments are all valuable and very helpful for revising and improving our paper, as well as the important guiding significance to our researches. We have studied comments carefully and have made correction which we hope meet with approval.

Improved portion are marked in red in the revised manuscripts. Because the most current version has added some words, this caused the line numbers has been changed. The main corrections in the paper and the responds to the editor’s and review’s comments are in following next page. Expect the editor to allow the request.

If you require any additional information regarding our manuscript, please do not hesitate to contact us directly via the resources below. Thank you for your time and consideration.

Thank you and best regards.

Sincerely,

Zhi-Ming Li

Heilongjiang Bayi Agricultural University, 163319, Daqing, China

E-mail: lizhiming1998@126.com

Responds to the reviewer’s comments:

  • Comment 1: Table 1: during the text, it must be reflected that there are no significant differences in liver, pancreas and cardiac indices.

Answer: Thank you for your comments. We have added this section as you suggested, please refer to lines 250-252 and 258-260 of the manuscript for details of the changes, which have been highlighted in red font.

  • Comment 2: I do not understand why the authors say that they cannot make the CRI, CRII and AIP indices if they have the individual values of TC, HDL, LDL and TG.

Answer: Thank you for your comments. We have added the Atherogenic Index of Atherogenic Index of Plasma (AIP), Castelli's Risk Index I (CRI-I) and Castelli's Risk Index II (CRI-II). For results of the presentation of the data, see Figure 3G H and I. In addition, we supplemented the manuscript with calculations of AIP, CRI-I, and CRI-II and analyzed the data results. Please see lines 159-161, 316-320, and 543-545 of the manuscript, respectively. We have marked the changes in red.

Figure 3. The effects of CHPs on lipid levels and liver function index of serum in T2DM mice (n=12). (A) TC, (B) TG, (C) HDL-c, (D) LDL-c, (E) ALT, (F) AST, (G) AIP, (H) CRI-I, (I) CRI-II. The data were presented as means ± S.D, which were analyzed by ANOVA test, followed by Tukey’s test between multiple groups. * indicates significant difference compared with the DC groups, *p < 0.05, **p <0.01, ***p < 0.001, ****p < 0.0001

  • Comment 3: Figure 4: indicate in the legend what the red arrow points to.

Answer: Thank you for your comments. We've explained what the red arrows in the diagram (Figure 4) point to. The additions are visible in lines 356-357 of the manuscript and have been marked in red font.
